# Secondary Radar Beacons for Local Ad-Hoc Autonomous Robot Localization Systems [note 1]

**DOI:** 10.3390/s19245484

**Published:** 2019-12-12

**Authors:** Martin Schütz, Tatiana Pavlenko, Martin Vossiek

**Affiliations:** Institute of Microwaves and Photonics, University of Erlangen-Nuremberg, 91058 Erlangen, Germany; tatiana.pavlenko@fau.de (T.P.); martin.vossiek@fau.de (M.V.)

**Keywords:** wireless local positioning, secondary radar, radar beacons, autonomous robot navigation, autonomous landing, space exploration

## Abstract

In this paper, we present a detailed analysis and implementation of secondary radar beacons designed for a local ad-hoc localization and landing system (LAOLa) to support the navigation of autonomous ground and aerial vehicles. We discuss a switched linear feedback network as a virtually coherent oscillator and show how to use it as a secondary radar transponder. Further, we present a signal model for the beat signal of the transponder response in an FMCW radar system, which is more detailed than in previously published papers. An actual transponder realization in the 24 GHz ISM band is presented. Its RF performance was evaluated both in the laboratory and in the field. Finally, we put forward some ideas on how to overcome the range measurement inaccuracy inherent in this transponder concept.

## 1. Introduction

Secondary radar systems have proven to be a reliable and performant solution for target localization and identification, especially in environments with weak or denied GNSS performance. They can be used for both indoor and outdoor localization, going down to centimeter precision and accuracy with update rates of more than 10 Hz. Fusing time-of-arrival (TOA), time-difference-of-arrival (TDOA), and angle-of-arrival (AOA) localization methods improves measurement precision and reliability in secondary radar setups with multiple stations [1,2,3,4,5].

A drawback of this system concept is that every participating radar station needs to provide a complex base band processing system for receive signal evaluation and transmit signal generation, even if a station acts only as a static reference. For the latter, a simple backscatter transponder with modulated response is acceptable and can be realized with much less effort. Several architectures for radar backscatter transponders, such as simple passive systems with modulated antenna base impedance [6], active transponders with heterodyne or homodyne down conversion [7], or active backscatters based on direct modulation of the RF signal, have been published in the past [8].

An alternative circuit for realizing active backscatter transponders is the super regenerative oscillator [9], whose internal output signal is radiated back over the receive antenna. By switching the oscillator on and off periodically, the signal appears as a virtually coherent response to the radar system. This concept was previously published as the switched-injection locked oscillator (SILO) in [10,11]. It is a simple but high-performance circuit for creating miniaturized, active backscatter transponders for radar localization systems [12]. The SILO was used for both localization and communication and a series of realizations for a great number of frequencies are available in the literature. Figure 1 summarizes a series of related publications [13,14,15,16,17,18,19,20,21].

Figure 2 shows the general secondary radar setup with a SILO-based backscatter transponder as published in the past [13,14]. The fact that transient oscillator behavior severely degrades the FMCW range measurement accuracy is ignored in prior publications. Further, exactly how the transponder response beat signal emerges in an FMCW radar system was not satisfactorily established.

We discuss the SILO as a linear feedback oscillator and derive the dependence of its oscillation amplitude and phase on the incidental radar signal in the following section. In Section 3, we show how to use this phase dependence to create virtually coherent, regenerative backscatter transponders for FMCW radar systems, and discuss some of the disadvantages of this system. A formally published SILO transponder system is presented in detail in Section 4, along with a lab characterization and field measurements. The transponder was used in our paper published at the 2019 IEEE International Workshop on Metrology for AeroSpace (MetroAeroSpace) [15]. In the last section, we outline the future trajectory of robot navigation and SILO-based backscatter transponders.

## 2. Switched Oscillators as Virtually Coherent Transponders

In this section, we review some fundamental equations relating to the super regenerative oscillator and consider, in particular, its sensitivity to the receiving signal phase. A good grounding in these fundamentals is needed to understand how this concept can be used to build coherent transponders for secondary radar applications in a straightforward manner.

### 2.1. The Linear Feedback Oscillator

As a basis for super regenerative receiving, we consider the linear feedback oscillator, which consists of an amplifier and a linear system F(s) in the feedback path as shown in Figure 3. The linear system is typically a bandpass filter at the desired oscillation frequency $\omega_{0}$ with bandwidth *B*. The oscillator output is given by (1)$$\begin{array}{c} s_{o}\left(t\right)=g\left(s_{o}\left(t\right)\;*\;f\left(t\right)+s_{i}\left(t\right)\right). \end{array}$$

This equation is transformed into the Laplace domain where it can be analyzed easily. After Laplace transformation, Equation (Equation 1) reads as follows: (2)$$\begin{array}{c} S_{o}\left(s\right)=g\left(S_{o}\left(s\right)\;\cdot \;F\left(s\right)+S_{i}\left(s\right)\right). \end{array}$$

By reordering according to $S_{o}\left(s\right)$, this leads to the well-known transfer function for feedback systems:(3)$$\begin{array}{c} S_{o}\left(s\right)=\frac{S_{i}\left(s\right)}{1-gF(s)}. \end{array}$$

Considering F(s) as a second-order bandpass filter with bandwidth $B_{f}$, center frequency $\omega_{0}$ and passband attenuation α, given by (4)$$\begin{array}{cc} F(s) & =\alpha \frac{2\pi B_{f}\frac{s}{\omega_{0}^{2}}}{\frac{s^{2}}{\omega_{0}^{2}}+2\pi B_{f}\frac{s}{\omega_{0}^{2}}+1}=\alpha \frac{Bs}{s^{2}+Bs+\omega_{0}^{2}},\;B=2\pi B_{f}, \end{array}$$

Equation (Equation 3) is written as (5)$$S_{o}\left(s\right)=\frac{S_{i}\left(s\right)}{1-\alpha g\frac{Bs}{s^{2}+Bs+\omega_{0}^{2}}}=S_{i}\left(s\right)\frac{s^{2}+Bs+\omega_{0}^{2}}{s^{2}+Bs\left(1-D\right)+\omega_{0}^{2}},\;D=\alpha g.$$

By exciting the feedback system with a Dirac impulse, i.e., $S_{i}\left(s\right)=1$, the impulse response is obtained:(6)$$\begin{array}{cc} S_{o}\left(s\right) & =\frac{s^{2}+Bs+\omega_{0}^{2}}{s^{2}+Bs\left(1-D\right)+\omega_{0}^{2}}\\ & =\frac{s^{2}}{s^{2}+Bs\left(1-D\right)+\omega_{0}^{2}}+\frac{Bs}{s^{2}+Bs\left(1-D\right)+\omega_{0}^{2}}+\frac{\omega_{0}^{2}}{s^{2}+Bs\left(1-D\right)+\omega_{0}^{2}}. \end{array}$$

The latter is transformed to the time domain by well-known transformation pairs. Considering D=1, Equation (Equation 6) is written as (7)$$\begin{array}{cc} S_{o}\left(s\right){|}_{D=1} & =\frac{s^{2}}{s^{2}+\omega_{0}^{2}}+\frac{Bs}{s^{2}+\omega_{0}^{2}}+\frac{\omega_{0}^{2}}{s^{2}+\omega_{0}^{2}}, \end{array}$$ which, after inverse Laplace transformation, is written in the time domain as (8)$$\begin{array}{cc} s_{o}\left(t\right){|}_{D=1} & =-\sin\left(\omega_{0}t\right)\varepsilon \left(t\right)+B\cos\left(\omega_{0}t\right)\varepsilon \left(t\right)+\sin\left(\omega_{0}t\right)\varepsilon \left(t\right)\\ & =B\cos\left(\omega_{0}t\right)\varepsilon \left(t\right). \end{array}$$

Equation (Equation 8) describes a stable oscillation at the center frequency of the bandpass filter. More interestingly, the oscillation amplitude is a function of the filter bandwidth. This is because more energy from the broad band Dirac impulse is injected into the system, which continues oscillating without damping.

For the general case D∈R, the amplitude of the oscillator output signal is either stable, rising, or damping, depending on the ratio of α and *g*: (9)$$\begin{array}{cc} S_{o}\left(s\right) & =\frac{s^{2}}{s^{2}+Bs\left(1-D\right)+\omega_{0}^{2}}+\frac{Bs}{s^{2}+Bs\left(1-D\right)+\omega_{0}^{2}}+\frac{\omega_{0}^{2}}{s^{2}+Bs\left(1-D\right)+\omega_{0}^{2}} \end{array}$$

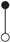

(10)$$\begin{array}{cc} s_{o}\left(t\right)=\; & \frac{e^{-B(1-D)t/2}}{\omega_{M}}\left(\left(\frac{1}{4}B^{2}{\left(1-D\right)}^{2}-\omega_{M}^{2}\right)\sin\left(\omega_{M}t\right)-B\left(1-D\right)\omega_{M}\cos\left(\omega_{M}t\right)\right)\\ & +\frac{B}{\omega_{M}}e^{-B(1-D)t/2}\left(\left(-\frac{B(1-D)}{2}\right)\sin\left(\omega_{M}t\right)+\omega_{M}\cos\left(\omega_{M}t\right)\right)\\ & +\frac{\omega_{0}^{2}}{\omega_{M}}e^{-B(1-D)t/2}\sin\left(\omega_{M}t\right)\\ & =\frac{e^{-B(1-D)t/2}}{\omega_{M}}\left(\left(\omega_{0}^{2}-\omega_{M}^{2}+\frac{1}{4}B^{2}{\left(1-D\right)}^{2}-\frac{1}{2}B\left(1-D\right)\right)\sin\left(\omega_{M}t\right)\right.\\ & \;\;+\omega_{M}\left.\left(1-B(1-D)\right)\cos\left(\omega_{M}t\right)\right)\\ & =e^{-B(1-D)t/2}\left(A_{\sin}\sin\left(\omega_{M}t\right)+A_{\cos}\cos\left(\omega_{M}t\right)\right). \end{array}$$

With the modified resonant frequency,
(11)$$\begin{array}{c} \omega_{M}=\sqrt{\omega_{0}^{2}-{\left(\frac{B(1-D)}{2}\right)}^{2}}. \end{array}$$

It is clear that, for D→1, Equation (Equation 10) converges to the undamped oscillation with fixed frequency described by Equation (Equation 8). This happens in practice when the amplifier is saturated and the oscillator reaches its maximum output power.

### 2.2. Signal-Injection Into the Linear Feedback Oscillator

In the above subsection, we examine the output signal of the linear feedback oscillator only for an exciting pulse, i.e., $S_{i}\left(s\right)=1$. Another interesting case is the excitation by a harmonic signal with a specific frequency $\omega_{i}$ and phase $\varphi_{i}$, which causes the oscillation to start. Here, the harmonic signal may force the feedback oscillator to its injecting frequency $\omega_{i}$ in the case of high input amplitude. In the case of weak input signals, they are just settling the starting condition for the oscillation (amplitude and phase). The latter, used as a receiver, is known as the super regenerative receiver, as its output signal depends directly on the phase and amplitude of the injected signal as show in Figure 4.

The Laplacian of a harmonic input signal with arbitrary frequency $\omega_{i}$, phase $\varphi_{i}$, and amplitude $A_{i}$ is given by (12)$$\begin{array}{cc} s_{i}\left(t\right) & =A_{i}\sin\left(\omega_{i}t+\varphi_{i}\right)\\ & =A_{i}\sin\left(\omega_{i}t\right)\cos\left(\varphi_{i}\right)+A_{i}\cos\left(\omega_{i}t\right)\sin\left(\varphi_{i}\right) \end{array}$$

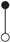

(13)$$\begin{array}{cc} S_{i}\left(s\right) & =A_{i}\frac{\omega_{i}}{s^{2}+\omega_{i}^{2}}\cos\left(\varphi_{i}\right)+A_{i}\frac{s}{s^{2}+\omega_{i}^{2}}\sin\left(\varphi_{i}\right). \end{array}$$

For this case, we reevaluate Equation (Equation 5), which yields
(14)$$\begin{array}{cc} S_{o,inj}\left(s\right)=\; & \frac{s^{2}+Bs+\omega_{0}^{2}}{s^{2}+Bs\left(1-D\right)+\omega_{0}^{2}}\;\;\cdot \;\;A_{i}\left(\frac{\omega_{i}}{s^{2}+\omega_{i}^{2}}\cos\left(\varphi_{i}\right)+\frac{s}{s^{2}+\omega_{i}^{2}}\sin\left(\varphi_{i}\right)\right). \end{array}$$

The inverse transformation reads as follows:(15)$$\begin{array}{cc} s_{o,inj}\left(t\right) & =A_{\mathrm{osc}}A_{i}e^{-B(1-D)t/2}\left(A_{\mathrm{osc},\sin}\left(\varphi_{i}\right)\sin\left(\omega_{M}t\right)+A_{\mathrm{osc},\cos}\left(\varphi_{i}\right)\cos\left(\omega_{M}t\right)\right)\\ & +A_{i}\left(A_{i,\cos}\left(\varphi_{i}\right)\cos\left(\omega_{i}t\right)+A_{i,\sin}\left(\varphi_{i}\right)\sin\left(\omega_{i}t\right)\right), \end{array}$$ again with the modified resonant frequency (16)$$\begin{array}{cc} \omega_{M} & =\sqrt{\omega_{0}^{2}-{\left(\frac{B(1-D)}{2}\right)}^{2}} \end{array}$$ and with the injection-phase dependant amplitudes $A_{\mathrm{osc},\sin}\left(\varphi_{i}\right),A_{\mathrm{osc},\cos}\left(\varphi_{i}\right),A_{i,\sin}\left(\varphi_{i}\right),A_{i,\cos}\left(\varphi_{i}\right)$, which can be approximated, as $\omega_{M}\approx \omega_{0}$ and $\omega_{0}\approx \omega_{i}$, with very good accuracy by (17)$$\begin{array}{cc} A_{\mathrm{osc}} & =\frac{1}{{\omega_{i}}^{4}+\left(-2{\omega_{0}}^{2}+B^{2}\;D^{2}-2B^{2}D+B^{2}\right){\omega_{i}}^{2}+{\omega_{0}}^{4}}\\ & =\frac{1}{\left(B^{2}D^{2}-2B^{2}D+B^{2}\right)\omega_{i}^{2}} \end{array}$$
(18)$$\begin{array}{cc} A_{\mathrm{osc},\sin}\left(\varphi_{i}\right) & =\left(\left(\left(-2BD\;{\omega_{0}}^{2}+B^{3}\;D^{3}-2B^{3}\;D^{2}+B^{3}D\right){\omega_{i}}^{2}+2BD\;{\omega_{0}}^{4}\right)\sin\left(\varphi_{i}\right)\right.\\ & +\left.\left(\left(B^{2}\;D^{2}-B^{2}D\right)\;{\omega_{i}}^{3}+\left(B^{2}\;D^{2}-B^{2}D\right)\;{\omega_{0}}^{2}\omega_{i}\right)\cos\left(\varphi_{i}\right)\right)\\ & \;/\left(\sqrt{4{\omega_{0}}^{2}-B^{2}\;D^{2}+2B^{2}D-B^{2}}\right)\\ & \approx \left(B^{2}\;D^{2}-B^{2}D\right)\;{\omega_{i}}^{2}\cos\left(\varphi_{i}\right)={\tilde{A}}_{\mathrm{osc}}\cos\left(\varphi_{i}\right) \end{array}$$
(19)$$\begin{array}{cc} A_{\mathrm{osc},\cos}\left(\varphi_{i}\right) & =\left(\left(B^{2}\;D^{2}-B^{2}D\right)\;{\omega_{i}}^{2}\sin\left(\varphi_{i}\right)+\left(BD\;{\omega_{i}}^{3}-BD\;{\omega_{0}}^{2}\omega_{i}\right)\cos\left(\varphi_{i}\right)\right)\\ & \approx \left(B^{2}\;D^{2}-B^{2}D\right)\;{\omega_{i}}^{2}\sin\left(\varphi_{i}\right)={\tilde{A}}_{\mathrm{osc}}\sin\left(\varphi_{i}\right) \end{array}$$
(20)$$\begin{array}{cc} A_{i,\sin}\left(\varphi_{i}\right) & =\frac{\left(BD\;{\omega_{i}}^{3}-BD\;{\omega_{0}}^{2}\omega_{i}\right)\sin\left(\varphi_{i}\right)+\left({\omega_{i}}^{4}+\left(-2{\omega_{0}}^{2}-B^{2}D+B^{2}\right){\omega_{i}}^{2}+{\omega_{0}}^{4}\right)\cos\left(\varphi_{i}\right)}{{\omega_{i}}^{4}+\left(-2{\omega_{0}}^{2}+B^{2}\;D^{2}-2B^{2}D+B^{2}\right){\omega_{i}}^{2}+{\omega_{0}}^{4}}\\ & \approx \cos(\varphi_{i}) \end{array}$$
(21)$$\begin{array}{cc} A_{i,\cos}\left(\varphi_{i}\right) & =\frac{\left(\left({\omega_{i}}^{4}+\left(-2{\omega_{0}}^{2}-B^{2}D+B^{2}\right){\omega_{i}}^{2}+{\omega_{0}}^{4}\right)\sin\left(\varphi_{i}\right)+\left(BD\;{\omega_{0}}^{2}\omega_{i}-BD\;{\omega_{i}}^{3}\right)\cos\left(\varphi_{i}\right)\right)}{{\omega_{i}}^{4}+\left(-2{\omega_{0}}^{2}+B^{2}\;D^{2}-2B^{2}D+B^{2}\right){\omega_{i}}^{2}+{\omega_{0}}^{4}}\\ & \approx \sin(\varphi_{i}) \end{array}$$

Based on the previous approximations, the solution of the time domain signal reads as follows:(22)$$\begin{array}{cc} s_{o,inj}\left(t\right) & \approx A_{\mathrm{osc}}{\tilde{A}}_{\mathrm{osc}}A_{i}e^{-B(1-D)t/2}\left(\cos\left(\varphi_{i}\right)\sin\left(\omega_{M}t\right)+\sin\left(\varphi_{i}\right)\cos\left(\omega_{M}t\right)\right)\\ & +A_{i}\left(\sin\left(\varphi_{i}\right)\cos\left(\omega_{i}t\right)+\cos\left(\varphi_{i}\right)\sin\left(\omega_{i}t\right)\right)\\ & ={\bar{A}}_{\mathrm{osc}}A_{i}e^{-B(1-D)t/2}\sin\left(\omega_{M}t+\varphi_{i}\right)+A_{i}\sin\left(\omega_{i}t+\varphi_{i}\right). \end{array}$$

Equation (Equation 22) shows a very important result for super regenerative receivers, namely, that the oscillation of the feedback linear oscillator depends on the injected signal’s amplitude and phase. In particular, the oscillation is coherent to the injected signal at the time of excitation t=0.

### 2.3. Simulation Results

To confirm the previous derivation, the time domain behavior of the switched linear feedback oscillator was simulated. We used a realistic parameter set, i.e.
$$\begin{array}{cc} f_{\mathrm{osc}} & =24.125\;\mathrm{GHz},\\ f_{\mathrm{inj}} & =24\;\mathrm{GHz},\\ B_{f} & =250\;\mathrm{MHz},\\ D & =10. \end{array}$$

Figure 5 shows the simulation results and, as expected, the oscillator is coherent to the injection signal at t=0. The oscillation amplitude quickly rises and stays virtually coherent to the radar signal. In real applications, the oscillator is switched off after a short period and switched on again after full decay.

## 3. Secondary Radar Beacons Based on Switched Oscillators

Based on the results from the previous section, we now show how the switched oscillator is used as a modulated backscatter transponder for secondary radar applications, also known as the switched-injection locked oscillator (SILO). For this, we review the SILO from [10] and derive a more sophisticated signal model for the case of an FMCW radar localization system. Here, we use several simplifications, which hold in real applications with very high accuracy.

### 3.1. Assumptions

We assume that the amplitude $A_{i}$ of the injected signal is negligible, i.e., (23)$$\begin{array}{cc} A_{i} & \ll {\bar{A}}_{\mathrm{osc}}A_{i}e^{-B(1-D)t/2},\;t>0,\\ \Leftrightarrow 1 & \ll {\bar{A}}_{\mathrm{osc}}e^{-B(1-D)t/2},\;t>0, \end{array}$$ which is reasonable in radar applications with mid to large ranges. Thus, we write the oscillator signal of Equation (Equation 22) as (24)$$\begin{array}{c} s_{o,\mathrm{inj}}\left(t\right)={\bar{A}}_{osc}A_{i}e^{-B(1-D)t/2}\;\cdot \;\sin\left(\omega_{M}t+\varphi_{i}\right). \end{array}$$

For the initial signal model, we even omit the exponential envelope, thus the oscillator output signal is modeled by (25)$$\begin{array}{c} s_{\mathrm{osc}}\left(t\right)=\sin\left(\omega_{M}t+\varphi_{i}\right). \end{array}$$

Further, we set (26)$$\begin{array}{c} \omega_{M}\approx \omega_{0}=\omega_{\mathrm{osc}} \end{array}$$
as for real applications, the center frequency of the oscillator varies only about a few MHz in the 24 GHz or 77 GHz operation bands.

### 3.2. The Beat Signal of a Single Oscillator Cycle

In the following section, we want to derive the beat signal of a SILO transponder in an FMCW radar during a single oscillator on-cycle, as shown in Figure 6. For simplicity, we use IQ signals without loss of generality.

An FMCW radar system transmits a linear chirp with rate $\mu =B_{f}/T$, where $B_{f}$ is the chirp bandwidth and *T* the sweep interval. The transmit signal is given by (27)$$\begin{array}{c} s_{\mathrm{tx}}\left(t\right)=\exp\left\{j\left(\omega_{\mathrm{tx}}+\pi \mu t\right)t\right\}. \end{array}$$

The transponder receives the radar signal transmitted at time $t_{0}$ after the delay time $\tau_{\mathrm{tx}}$ and is switched on at time t. Without loss of generality, we set $t_{0}+\tau_{\mathrm{tx}}=t$. The oscillator turns on with the phase of the received radar signal, the injected phase (28)$$\begin{array}{cc} \varphi_{\mathrm{inj}}\left(t_{\mathrm{on}}\right) & =\arg\{s_{\mathrm{tx}}\left(t_{\mathrm{on}}-\tau_{\mathrm{tx}}\right)\}\\ & =\arg\{s_{\mathrm{tx}}\left(t_{0}\right)\}\\ & =\left(\omega_{\mathrm{tx}}+\pi \mu t_{0}\right)t_{0}. \end{array}$$

Thus, the SILO transmit signal during a single on-cycle with duration $T_{s}$ is given by Equation (Equation 25) (29)$$\begin{array}{ll} s_{\mathrm{osc}}\left(t\right) & =\exp\{j\omega_{\mathrm{osc}}\left(t-t_{\mathrm{on}}\right)+j\varphi_{\mathrm{inj}}\left(t_{\mathrm{on}}\right)\}\\ & =\exp\{j\omega_{\mathrm{osc}}\left(t-t_{\mathrm{on}}\right)+j\left(\omega_{\mathrm{tx}}+\pi \mu t_{0}\right)t_{0}\}\\ t & \in [t_{\mathrm{on}};t_{\mathrm{off}}]. \end{array}$$

The transponder signal $s_{\mathrm{osc}}\left(t\right)$ is delayed by $\tau_{\mathrm{rx}}$ and received by the radar system
(30)$$\begin{array}{lll} s_{\mathrm{rx}}\left(t\right) & = & s_{\mathrm{osc}}\left(t-\tau_{\mathrm{rx}}\right)\\ & = & \exp\{j\omega_{\mathrm{osc}}\left(t-t_{\mathrm{on}}-\tau_{\mathrm{rx}}\right)+j\left(\omega_{\mathrm{tx}}+\pi \mu t_{0}\right)t_{0}\}\\ t & \in & [t_{\mathrm{rx}};t_{\mathrm{rx}}+T_{s}]. \end{array}$$

In the FMCW receiver, $s_{\mathrm{rx}}\left(t\right)$ is down-converted with the local oscillator signal $s_{\mathrm{tx}}\left(t\right)$
(31)$$\begin{array}{lll} s_{b}\left(t\right) & = & s_{\mathrm{tx}}\left(t\right)\;\cdot \;s_{\mathrm{rx}}^{*}\left(t\right)\\ & = & \exp\{j\left(\omega_{\mathrm{tx}}+\pi \mu t\right)t-j\omega_{\mathrm{osc}}\left(t-t_{\mathrm{on}}-\tau_{\mathrm{rx}}\right)-\left(\omega_{\mathrm{tx}}+\pi \mu t_{0}\right)t_{0}\}\\ t & \in & [t_{\mathrm{rx}};t_{\mathrm{rx}}+T_{s}]. \end{array}$$

Equation (Equation 31) is the beat signal model of a single oscillator on-cycle within the FMCW sweep. At the time of reception $t_{\mathrm{rx}}=t_{0}+\tau_{\mathrm{tx}}+\tau_{\mathrm{rx}}=t_{0}+\tau_{\mathrm{rt}}$, this evaluates to (32)$$\begin{array}{cc} s_{b}\left(t_{\mathrm{rx}}\right) & =s_{b}\left(t_{0}+\tau_{\mathrm{rt}}\right)\\ & =\exp\{j\left(\omega_{\mathrm{tx}}+\pi \mu \left(t_{0}+\tau_{\mathrm{rt}}\right)\right)\left(t_{0}+\tau_{\mathrm{rt}}\right)-j\omega_{\mathrm{osc}}\left(t_{0}+\tau_{\mathrm{rt}}-t_{\mathrm{on}}-\tau_{\mathrm{rx}}\right)-j\left(\omega_{\mathrm{tx}}+\pi \mu t_{0}\right)t_{0}\}\\ & =\exp\{j\left(\omega_{\mathrm{tx}}t_{0}+\omega_{\mathrm{tx}}\tau_{\mathrm{rt}}+\mu \pi \left(t_{0}^{2}+2t_{0}\tau_{\mathrm{rt}}+\tau_{\mathrm{rt}}^{2}\right)-\omega_{\mathrm{tx}}t_{0}-\pi \mu t_{0}^{2}\right)\}\\ & =\exp\{j\left(2\pi \mu \tau_{\mathrm{rt}}t_{0}+\omega_{\mathrm{tx}}\tau_{\mathrm{rt}}+\pi \mu \tau_{\mathrm{rt}}^{2}\right)\}\\ & \approx \exp\{j\left(2\pi \mu \tau_{\mathrm{rt}}t_{0}+\omega_{\mathrm{tx}}\tau_{\mathrm{rt}}\right)\}. \end{array}$$

Equation (Equation 32) shows that the beat signal of the SILO transponder at $t_{\mathrm{rx}}=t_{0}+\tau_{\mathrm{rt}}$ has the same phase as a signal of a passive scatterer with the same round-trip delay time $\tau_{\mathrm{rt}}$. Within the oscillator on-time $T_{s}$, a specific bandwidth $\mu T_{s}$ is traversed:(33)$$\begin{array}{ll} s_{b}\left(t_{\mathrm{rx}}+T_{s}\right) & =s_{b}\left(t_{0}+\left(\tau_{\mathrm{rt}}+T_{s}\right)\right)\\ & =\exp\{j\left(2\pi \mu \left(\tau_{\mathrm{rt}}+T_{s}\right)t_{0}+\omega_{\mathrm{tx}}\left(\tau_{\mathrm{rt}}+T_{s}\right)+\pi \mu {\left(\tau_{\mathrm{rt}}+T_{s}\right)}^{2}-\omega_{\mathrm{osc}}T_{s}\right)\}\\ & \approx s_{b}\left(t_{0}+\tau_{\mathrm{rt}}\right)\;\cdot \;\exp\{j\left(2\pi \mu T_{s}t_{0}+\omega_{\mathrm{tx}}T_{s}-\omega_{\mathrm{osc}}T_{s}\right)\\ & =s_{b}\left(t_{0}+\tau_{\mathrm{rt}}\right)\;\cdot \;\exp\{j\left(2\pi \mu T_{s}t_{0}+\Delta \omega T_{s}\right). \end{array}$$

It also shows up that the beat signal’s phase is a function of the oscillator on-time $T_{s}$ and of the offset Δω of the transmit frequency $\omega_{\mathrm{tx}}$ and oscillator frequency $\omega_{\mathrm{osc}}$.

### 3.3. The Beat Signal of the Entire Frequency Sweep

Equations (Equation 32) and (Equation 33) describe the beat signal during a single oscillator on-cycle. Within the entire frequency sweep, the beat signal consists of multiple on-cycles at discrete times $t_{i}$, where in each cycle the signal is equal to a passive scatterer for the singular moment $t_{\mathrm{rx}}$ and induces an additional frequency bandwidth $2\pi \mu T_{s}$ until the oscillator is switched off. This is interpreted as a discrete time sampling process of a frequency band with bandwidth $2\pi \mu T_{s}$, starting at the actual beat frequency $2\pi \mu \tau_{\mathrm{rt}}$ with phase $\omega_{\mathrm{tx}}\tau_{\mathrm{rt}}$ and stopping with an additional phase $\Delta \omega T_{s}$. Spectral replicas, which need to be filtered in the baseband, appear at frequencies $1/T_{s}$ due to the discrete time sampling. During the oscillator on-time, the beat signal is a chirp starting from the beat-frequency $2\pi \mu \tau_{\mathrm{rt}}$ with bandwidth $2\pi \mu T_{s}$. This chirp is approximated in the frequency domain with good accuracy by a rect-function with the same bandwidth. The reconstruction filter for the sampled signal is a lowpass filter with a cutoff frequency of at least the on/off frequency $1/T_{s}$. This is normally realized in real FMCW systems by the anti-aliasing filter before the analog-to-digital conversion stage.

As a result, the frequency spectrum of the beat signal of a SILO transponder within a linear FMCW ramp is written (with the window function W(ω)) as:(34)$$\begin{array}{cc} S_{b}\left(\omega \right) & =W_{T}\left(\omega \right)*\mathrm{rect}\left(\frac{\omega -2\pi \mu \tau_{\mathrm{rt}}-2\pi \mu T_{s}/2}{2\pi \mu T_{s}}\right)\;\cdot \;\exp\left\{j\left(\omega_{\mathrm{tx}}\tau_{\mathrm{rt}}+\Delta \omega T_{s}\frac{\omega -2\pi \mu \tau_{\mathrm{rt}}}{2\pi \mu T_{s}}\right)\right\}\\ & =W_{T}\left(\omega \right)*\mathrm{rect}\left(\frac{\omega }{2\pi \mu T_{s}}\right)e^{j\omega \frac{\Delta \omega }{2\pi \mu }}*\delta \left(\omega -2\pi \mu \tau_{\mathrm{rt}}-2\pi \mu T_{s}/2\right)e^{j\omega_{\mathrm{tx}}\tau_{\mathrm{rt}}-j\Delta \omega \tau_{\mathrm{rt}}}. \end{array}$$

The following time-domain model is obtained from Equation (Equation 34) (considering $\omega_{\mathrm{tx}}\tau_{\mathrm{rt}}-\Delta \omega \tau_{\mathrm{rt}}=\omega_{\mathrm{osc}}\tau_{\mathrm{rt}}$):(35)$$\begin{array}{cc} s_{b}\left(t\right)=\; & w_{T}\left(t\right)\;\cdot \;\mathrm{sinc}\left(\frac{2\pi \mu T_{s}\left(t-\frac{\Delta \omega }{2\pi \mu }\right)}{2}\right)\;\cdot \;\exp\left\{j2\pi \left(\mu \tau_{\mathrm{rt}}+\frac{\mu T_{s}}{2}\right)t+j\omega_{\mathrm{osc}}\tau_{\mathrm{rt}})\right\} \end{array}$$

For simplicity, all amplitude values are normalized to 1 as they carry no relevant information.

If we assume that $\omega_{\mathrm{osc}}$ is in the middle of the operating band (what is preferable for real applications) and if t∈[0;T], the delta Δω from the oscillator frequency to the FMCW start frequency is given by:(36)$$\Delta \omega =2\pi \mu \frac{T}{2}.$$

By modulating the amplitude of the transponder output signal, either by on/off keying or by controlling the output power of the oscillator with a periodic function with frequency $\omega_{m}$, the beat spectrum is shifted to an arbitrary center frequency. This allows the reader to distinguish the transponder signal clearly from primary radar targets and clutter. Thus, the beat signal is written as (37)$$\begin{array}{cc} s_{b}\left(t\right)= & w_{T}\left(t\right)\;\cdot \;\mathrm{sinc}\left(\frac{2\pi \mu T_{s}\left(t-T/2\right)}{2}\right)\;\cdot \;\exp\left\{j2\pi \left(\mu \tau_{\mathrm{rt}}+\frac{\mu T_{s}}{2}\right)t+j\omega_{\mathrm{osc}}\tau_{\mathrm{rt}})\right\}\;\cdot \;\cos\left(\omega_{m}t+\varphi_{m}\right). \end{array}$$

Figure 7 shows the measured beat frequency spectrum of a real transponder with modulated reflection.

### 3.4. Measuring the Absolute Distance to a SILO Transponder

By transforming a measured beat signal into the frequency domain, the signal round trip time $\tau_{\mathrm{rt}}$ and thus the transponder distance *d* is measured by the frequency difference between the two AM sidebands, as shown in Figure 7:(38)$$\begin{array}{c} d_{\omega }=c\frac{\Delta \omega_{R}}{4\mu }-c\frac{T_{s}}{4},\;\Delta \omega_{R}=\omega_{2}-\omega_{1}. \end{array}$$

In practical systems, the correction term $c\frac{T_{s}}{4}$ depends on the transient behavior of the oscillator and cannot be calibrated accurately, because the latter is a function of the temperature and especially of the strength of the injected power, which depends on the other hand on the transponder distance and on the orientation because of the antenna pattern. It is good practice to calibrate the distance error at the middle of the operational range and accept the range inaccuracy. Below, in the Outlook Section 6, we present some ideas on how to include the range error correction term in a multi-static radar setup.

## 4. Realization of a Miniaturized Secondary Radar Beacon

In this section, we give a detailed description of a previously presented, miniaturized secondary radar transponder based on the switched injection locked oscillator in the 24 GHz ISM-Band. The transponder comprises of a four-layer PCB, whose dimensions are 35 mm×15 mm and which uses a standard FR4 combined with Rogers RO4003 as a layer stack. The Rogers substrate exhibits high RF performance because of its homogeneous electrical permittivity and low dielectric losses; it is used to realize the microwave oscillator in microstrip technology.

### 4.1. Overview

Figure 8 gives an overview of the essential devices on the top and bottom layers of the transponder hardware. The actual switched-injection locked oscillator, consisting of a feedback amplifier in the form of a microwave field-effect transistor of type CEL CE3520K3, is located on the top layer. The SILO is described in detail in Section 4.2. The RF signals are fed over an SMP connection from and into the system. SMP provides a low-cost but performant RF link at 24 GHz without the need for bolting. This is a small footprint feature. A chip of type MAX8663 provides power management for the transponder. It converts the primary voltages to the target voltages. It also maintains a balance between the USB power supply and the battery, whereby the latter is charged automatically when there is sufficient USB power. The RF oscillator voltage is produced apart from the peripheral supply because it needs a high degree of linearity along with low noise. For this, a linear regulator of type LT3042 is used, which provides a high PSRR of about 80 dB between 0 and 2 MHz. A good PSRR number is crucial, as any amplitude modulation on the power supply will create unexpected sidebands in the RF oscillation and thus ghost targets in the radar base band. Furthermore, the power supply noise affects the oscillator’s spectral purity and its sensitivity to the receive signal’s phase, which is directly related to the SNR of the transponder response in the base band. Finally, there are minor devices on the top layer such as a balun to create a single-ended modulation signal out of the symmetric DDS output.

All transponder digital periphery is located on the bottom layer. An STM32L100 microcontroller and an AD9102 DDS contain the transponder response modulation IP. Further, the STM32L100 provides a USB interface for online modulation parameter configuration. The transponder can be powered either by USB or by a LiPo battery, connected through a PHR2 socket.

### 4.2. The SILO Core

The switched-injection locked oscillator is built, as shown in Figure 9, of an amplifier with positive feedback at the target frequency, which is the center of the 24 GHz ISM band in this case. The amplifier is realized by a GaAs RF FET of type CE3520K3 as a grounded-emitter circuit. The latter provides the highest power amplification among transistor circuits and is convenient for realizing the oscillator.

The periodic on/off switching action is realized by push/pulling the oscillator’s supply voltage. The push/pull operation is implemented by a second p-channel FET. By pulling the supply voltage, the oscillator starves and becomes ready for the next on-cycle very quickly. Thus, a fast switching frequency can be realized by maintaining the phase sensitivity at the same time. The transistor gate voltage is defined at the switch-on time solely by the injected signal and thermal noise. By pushing the supply voltage, the loop gain becomes positive and the oscillation starts with the receive signal phase, as explained in Section 2. The oscillation is radiated over the antenna and the oscillator is switched off again after a short period.

## 5. Evaluation of the Transponder

The transponder was tested and assessed both in the lab and in the field, to characterize its performance and test its operation as a secondary radar beacon. First, the transponder was tested *in the loop*, i.e., without antennas, but with a separate signal source to produce the injecting signal, and a high-speed oscilloscope to verify the switching and locking performance. Further, the circuit sensitivity, i.e., the minimum power needed for injection locking, was measured. The transponders were then tested with antennas, but without signal injection and without switching. The steady-state output power and center frequency can be measured easily in this setup. The transponder was tested finally with a real FMCW radar unit in order to verify its function as a radar beacon. The response spectrum was examined in this context and a real distance measurement was made and compared with total station reference data.

### 5.1. Hardware in the Loop

The transponder was evaluated in the lab with the setup shown in Figure 10. The injection signal was generated by a Rohde & Schwarz SMF100A signal generator, and a high-speed Agilent DSO-X 925504A oscilloscope was connected to the transponder to measure both the injection signal and the transponder response. All three devices were put together through an Agilent 11667C power splitter. The minimum output power of the signal generator is −20 dBm. Series of attenuators were connected before the power splitter in order to reduce the input power.

#### 5.1.1. Evaluation of the Free-Running, Non-Switched Oscillator

The transponder was connected to the measurement setup and operated without modulation, only with the oscillator fixed power supply. Thus, the output power and the center frequency can be measured easily. Further, possible spurious signals can be identified and characterized. Figure 11a shows the amplitude spectrum of the measured output signal. There is a stable oscillation at approximately 24.165GHz with 5 dBm power and within the total spectrum there is no interference higher than −70 dBc. Figure 11b shows a detailed view of the transponder spectrum, which gives the oscillator output power and center frequency. The spectrum was calculated from only a few samples to mitigate the oscillator drift.

#### 5.1.2. Evaluation of the Free-Running, Switched Oscillator

Figure 12a shows the amplitude spectrum of the output signal for a switched oscillator, but without signal injection. It appears that there is no oscillation at a fixed frequency, but a raised noise floor around the oscillator center frequency. This is caused by the fact that, for every oscillator on-cycle, the oscillation is coherent to a random phase and, thus the overall output signal is a band-limited noise signal. It shows the frequency transfer function of the feedback loop and the frequency dependence of the loop gain. An important result is that no self-locking appears, which means that the oscillation of one on-cycle does not disturb the phase of the successive cycle, which gives the SILO its maximum sensitivity.

#### 5.1.3. Evaluation of the Free-Running, Switched Oscillation with Signal Injection

The last evaluation in the lab was the signal injection into the switched oscillator in order to test the coherent oscillation. Since the signal source has a minimum power level of −20 dBm, the signal was damped with a series of attenuators to set the injection power down to −80 dBm. The output spectrum for injection locking is shown in Figure 12c. The oscillator locks virtually to the frequency of the injected signal, because the phase in each on-cycle is coherent at t=0. Sidebands appear at multiples of the switching frequency (90 MHz) because of the periodic on-off-switching. The SILO sensitivity, where injection locking is barely seen, was evaluated by decreasing the injected power. It is evident that the injection locking vanishes for signals below approximately −70 dBm and the noise output, as shown in Figure 12a, remains.

The time domain output of a single oscillator on-cycle is shown for different injection power levels in Figure 13. In the latter, the coherent start of the oscillation is clearly seen. It also proves that the SILO pulse form depends on the input power, which becomes important for localization applications.

### 5.2. The Free-Running, Switched and Modulated Transponder Without Injection

In the next step, the transponder hardware was equipped with an antenna, as it is used in the real localization scenario. Additionally, a second antenna was connected to the oscilloscope to measure the transponder radiated signal. Mismatches in the RF and antenna hook-ups, which may cause center frequency deviations or even total oscillator malfunction, can thus be detected. Further, in this setup, the low frequency amplitude modulation of the output signal was configured and measured. This is crucial for setting the spectral position of the transponder signal in the radar base band. Figure 14 shows the measured output signal in the time domain. Another option is to use on–off-keying as low frequency modulation and maximum signal power in the fundamental frequency, but it would create harmonics in the radar base band.

### 5.3. Test with a Real Radar System

The transponder was tested in a real distance measurement scenario using the radar system that was built for the LAOLa project. Reference data were provided by a Leica TS30 total station in tracking mode. Figure 15a shows the range spectrum of the modulated response and Figure 15b the distance measured with the radar system compared to the total station. There is very good accordance between both plots, with the measurement update rate of the radar system being superior to that of the total station. The outlier at sample 120 might have been caused by a multipath distortion, e.g., by a ground reflection. The distance error at the center of the operational range was used as the distance error correction term, and it was assumed to be constant over the entire range. The distance error is shown in Figure 15c with an RMSE of 5.64cm.

## 6. Outlook

### 6.1. The SILO Transponder in a Cooperative Radar Setup

As mentioned, the transponder concept presented in this paper introduces a range uncertainty that is a function of the oscillator’s transient behavior. In general, this error cannot be measured with a single radar channel, because it is independent of all radar parameters such as the ramp duration or bandwidth.

However, by introducing a second receive channel or an additional, cooperative radar system, respectively, the difference of the time of arrival/distance can be calculated:(39)$$\begin{array}{cc} \Delta d=d_{1}-d_{2} & =c\frac{\Delta \omega_{R,1}}{4\mu }-c\frac{T_{s}}{4}-c\frac{\Delta \omega_{R,2}}{4\mu }+c\frac{T_{s}}{4}\\ & =c\frac{\Delta \omega_{R,1}}{4\mu }-d_{\mathrm{err}}-c\frac{\Delta \omega_{R,2}}{4\mu }+d_{\mathrm{err}}\\ & =c\frac{\Delta \omega_{R,12}}{4\mu } \end{array}$$ which results again in an accurate measurement without the influence of the transient behavior of the oscillator. With Δd and the positions of the radar sensors, a solution for the absolute distance can be found by solving the general TDOA equation [22].

By comparing the latter solution to the direct measurements, an estimate for the error is obtained (40)$$\begin{array}{c} {\hat{d}}_{\mathrm{err}}=d_{1}-d_{\mathrm{TDOA}} \end{array}$$

This estimate can be computed, e.g., recursively in a Kalman filter during the measurements, whereby the filter output will have the accuracy of the TDOA evaluation and the precision of the TOA evaluation. A Kalman filter that relies only on TDOA measurements can be deployed as well for localization [23].

### 6.2. Active Backscatter Transponders for Automotive Radar Systems

There is still a gap in the frequency range for which SILO transponders have been realized. There are no valuable transponders available at this time—especially for the 77 GHz automotive band. It would provide a great opportunity for building low-cost radar target simulators with high RCS for radar calibration and verification [6].

## 7. Conclusions

Switched-injection locked oscillators (SILOs) provide a simple but powerful circuit for creating active radar transponders with modulated backscatter. They are used as secondary radar beacons in local ad-hoc localization systems for autonomous robot navigation. The phase coherence of the SILO to the radar signal and a corresponding FMCW beat signal of the response are derived based on a liner system model. A working system and its performance for the 24 GHz ISM band are shown. Because of the transient behavior of the switched oscillator, a range uncertainty is introduced, which needs to be calibrated or considered, and estimated online in the overall localization algorithm.

## Figures and Tables

**Figure 1 sensors-19-05484-f001:**
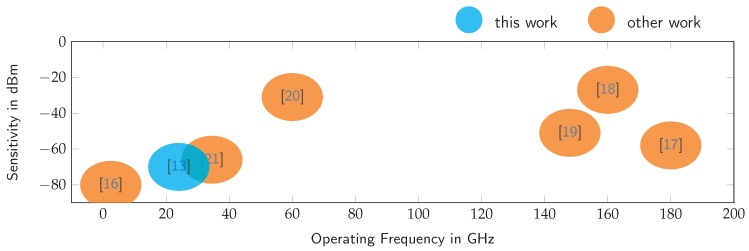
Previously published implementations of the switched-injection locked oscillator.

**Figure 2 sensors-19-05484-f002:**
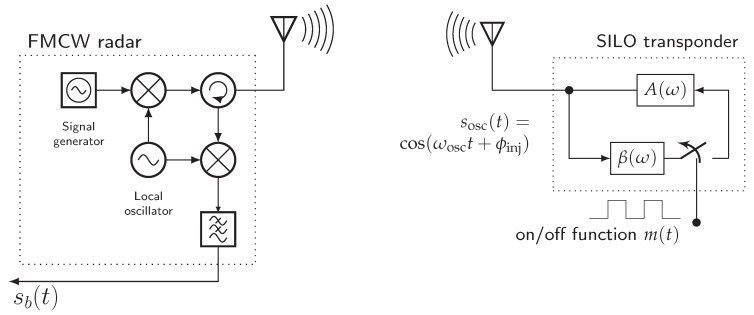
The general setup of a switched injection-locked transponder and an FMCW radar system.

**Figure 3 sensors-19-05484-f003:**
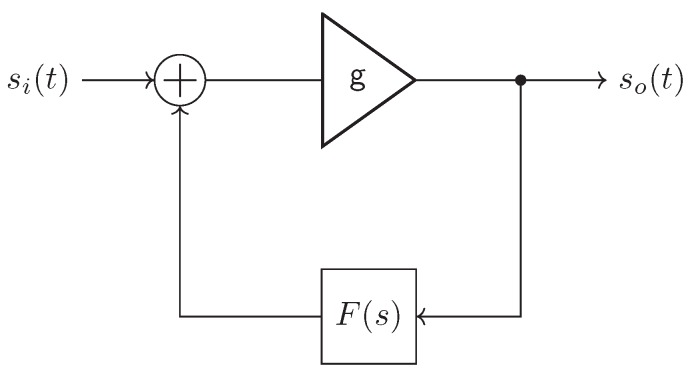
The input–output relationship of the linear feedback oscillator.

**Figure 4 sensors-19-05484-f004:**
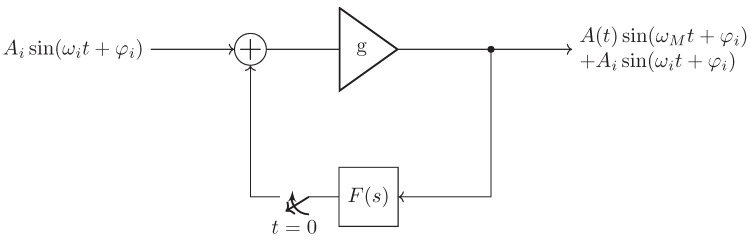
By exciting the linear feedback oscillator with a harmonic signal, the phase of the output signal is coherent to the input at t=0.

**Figure 5 sensors-19-05484-f005:**
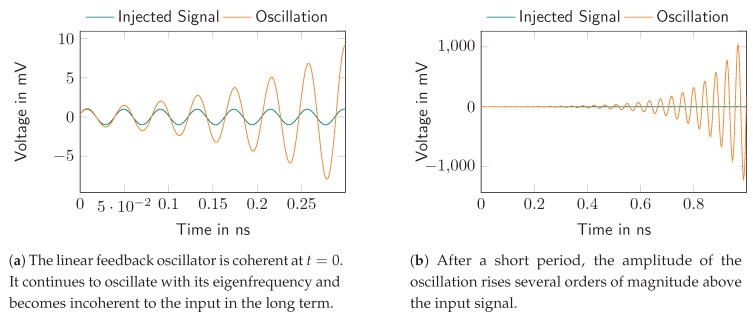
Simulation results for the linear feedback oscillator with harmonic signal injection.

**Figure 6 sensors-19-05484-f006:**
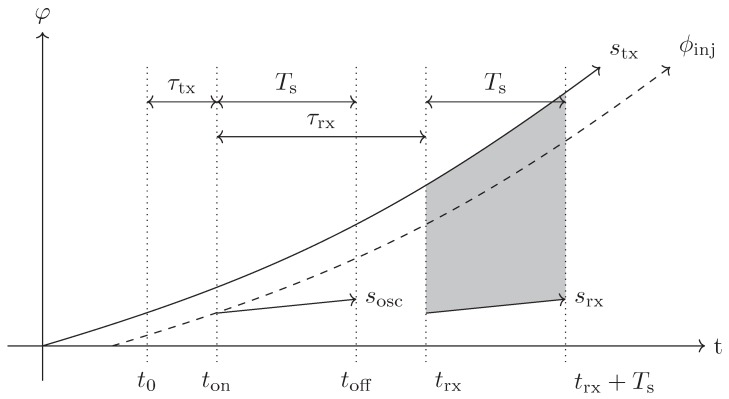
The instantaneous phases and induced bandwidth in the FMCW base band within one oscillator on-cycle.

**Figure 7 sensors-19-05484-f007:**
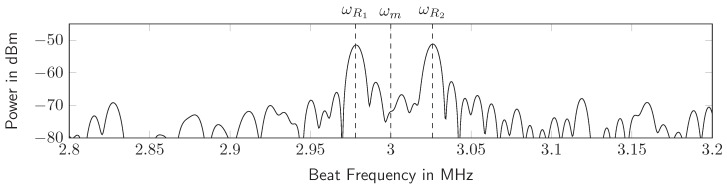
The transponder signal spectrum in the radar baseband. The center frequency $\omega_{m}$ is determined by the amplitude modulation of the transponder signal.

**Figure 8 sensors-19-05484-f008:**
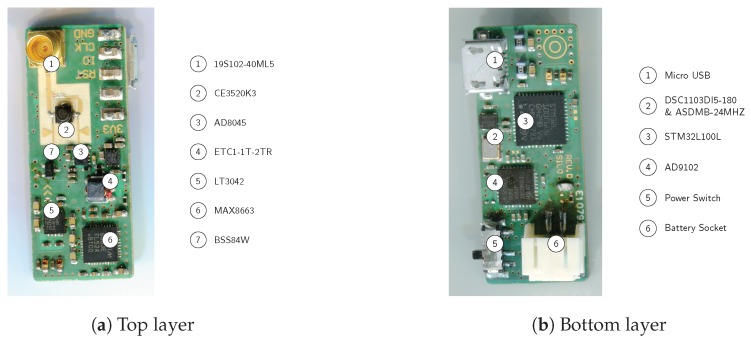
The most important parts of the transponder on the top and bottom layers

**Figure 9 sensors-19-05484-f009:**
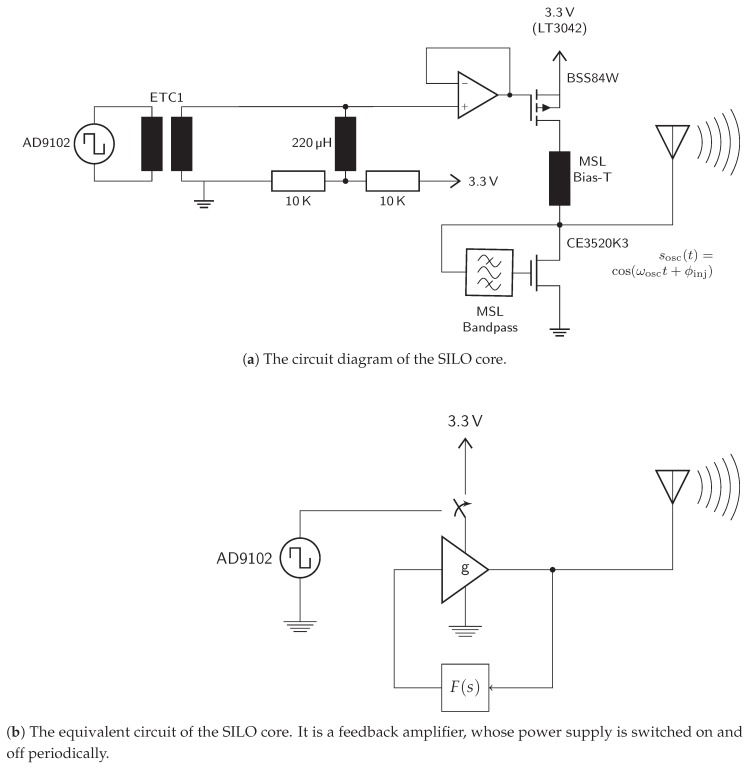
The switched injection locked oscillator core of the transponder. The modulation signal is generated by direct digital synthesis.

**Figure 10 sensors-19-05484-f010:**
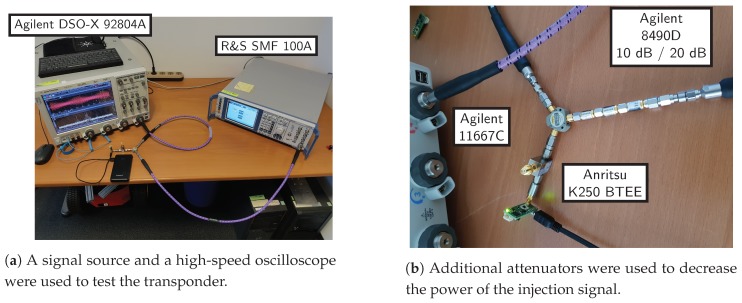
The setup for measuring the transponder hardware in the loop.

**Figure 11 sensors-19-05484-f011:**
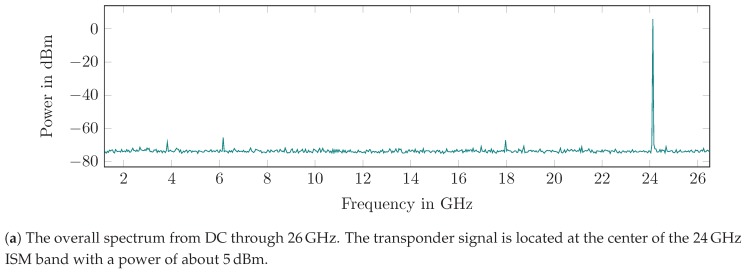
The spectrum of the free-running, non-switched oscillator.

**Figure 12 sensors-19-05484-f012:**
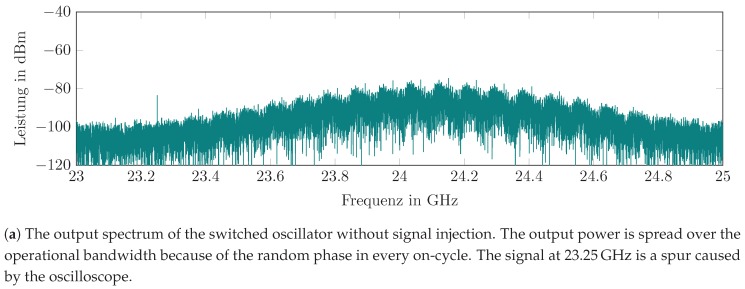
The evaluation of the oscillator output signal for different injection power levels.

**Figure 13 sensors-19-05484-f013:**
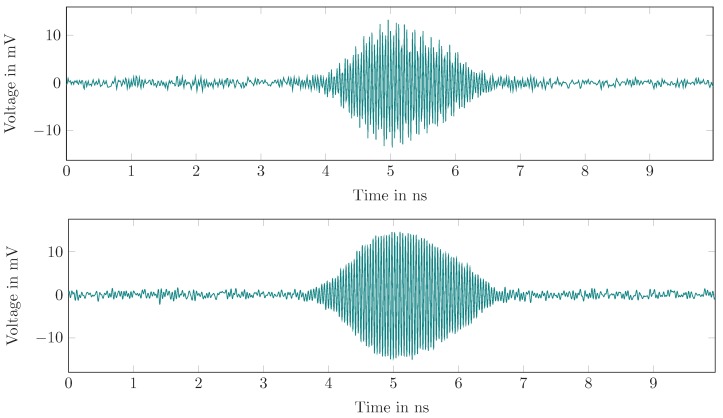
The evaluation of a single oscillator pulse for different injection power levels.

**Figure 14 sensors-19-05484-f014:**
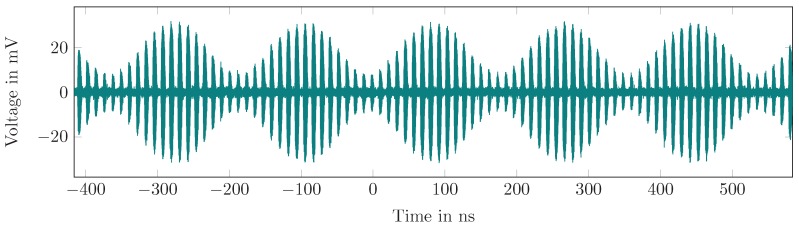
The modulated RF output of the transponder. It shows the high-frequency on–off switching and the low frequency amplitude modulation as explained in Equation (Equation 37).

**Figure 15 sensors-19-05484-f015:**
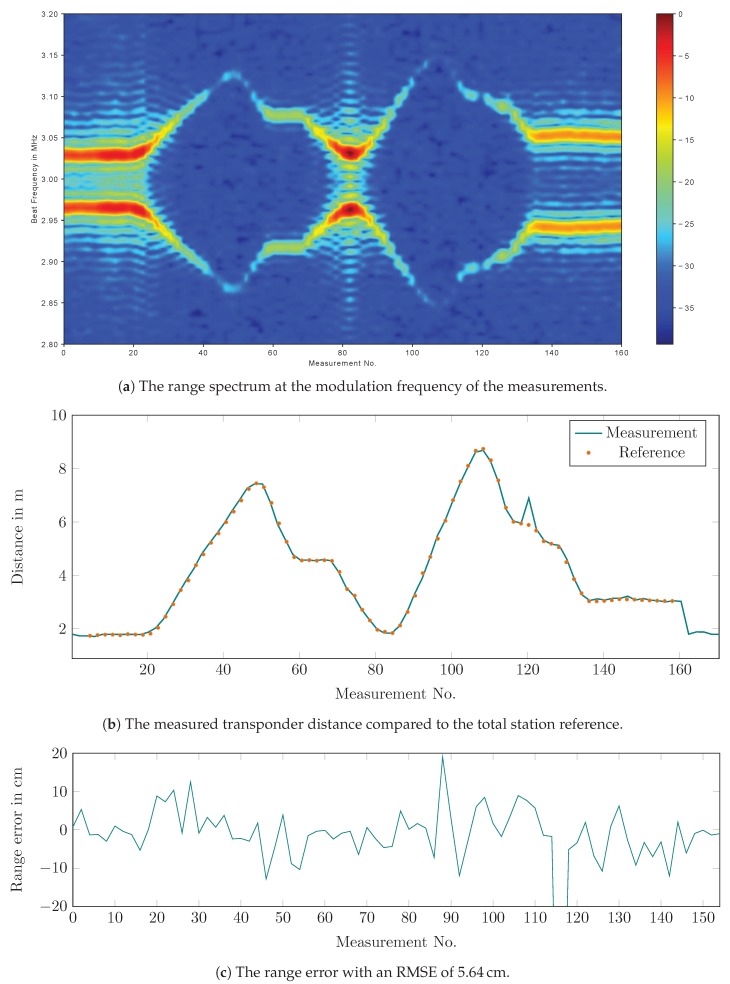
The transponder test with a real radar system.

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
