# Peer review of "Secondary Radar Beacons for Local Ad-Hoc Autonomous Robot Localization Systemsâ€"

_sensors, 2019, doi:10.3390/s19245484_

Round 1
Reviewer 1 Report
This paper proposes a more complex signal model for transponder response.However, in order for the reader to quickly grasp your contributions, it is best to highlight the main difficulties, challenges and background in the introduction section, as well as the original achievements in overcoming them.
Author Response
First of all, I would like to thank you for your time to review my paper.
The main difference in this publication is the consideration of the transient behavior of the switched oscillator and the exact description of the transponder response within an FMCW radar system. In my opinion, this was not considered in any of the previous publications about this architecture.
I will try to clear this in the final publication.
Best Regards
Martin

Reviewer 2 Report
The authors provided a work on the description of a secondary radar beacons developped for the LAOLa system. In general, the paper is prepared attentively. However, comments and suggestions below would make the paper higher quality.
Major comments:
-I think the abstract should be revised: it is not relevant in the abstract that this work is a detailed description of the work presented at MetroAerosapace 2019, this should be disclosed in the introduction section.
- I think the advantage of the proposed method over the state of the art is not clearly described in the introduction.
- I think that the authors shall include a histogram of the errors between the measured transponder distance compared to the total station reference and calculate the standard deviation. Moreover Fig 15(b) does not allow the reader to quantify the errors order of magnitude (cm or mm ?)
- Some experimental results of Section 6.1 shall be included.
Minor comments
- Fig. 2 is listed in the test before Fig. 1.
- All the acronym shall be specified: as an example FMCW
- I believe the transition from Eq. 6 to Eq. 7 is true only if α = 1/g and alpha = g is not correct. Moreover the motivation behind considering α = 1/g should be described in more detail.
-Eq 13: Si(s) and not Si(t).
- I think that the detail “The inverse transformation was calculated with the computer algebra system Maxima because of the expensive time domain representation” is not interesting for the reader.
- The transition from Eq. (17) to Eq. (18) is verified if ω_0 = ω_i, however it is not clearly disclosed in the text. Does it come from the assumption α= 1/g?
- Eq. (20) sin(φ) instead sin(p)
- Eq. (22) Aosc is introduced for the first time.
-Line 68: Isn’t the output of a feedback linear oscillator dependent on both amplitude and phase of the injected signal true for every feedback linear system? It does not surprise me.
- Fig. 6 is not listed in the text.
- Line 178: The reference to the figure is lost.
- Line 211: The reference to the figure is wrong.
Author Response
First of all, I would like to thank you for your time to review my paper.
The main difference in this paper is the consideration of the transient behavior of the switched oscillator and the exact description of the transponder response within an FMCW radar system. In my opinion, this was not considered in any of the previous publications about this architecture. I will try to clear this in the final publication.
I will insert an error history, but I don't have any measurements for the method described in section 6.1, may I can include some simulations.
I will fix the minor issues to round out the paper.
Best Regards
Martin

Round 2
Reviewer 2 Report
I went through this paper and the system developed is technically sound to me. Also, the authors have carefully revised the paper according to the reviewers' comments. The contribution of this paper with respect to the state of the art have been clearly disclosed in the introduction.
The research topic of the manuscript is very meaningful and interesting. The construction of the paper is proper. I find this paper of a very high quality and I recommend publishing.
Author Response
Thank you for you kind comments, I will do some English editing before submitting the final version.